# The Shuttling of Methyl Groups Between Folate and Choline Pathways

**DOI:** 10.3390/nu17152495

**Published:** 2025-07-30

**Authors:** Jonathan Bortz, Rima Obeid

**Affiliations:** 1Human Nutrition and Health, Balchem Corporation, 5 Paragon Drive, Montvale, NJ 07645, USA; jbortz@balchem.com; 2Department of Clinical Chemistry and Laboratory Medicine, Saarland University Hospital, D-66424 Homburg, Germany

**Keywords:** betaine, choline, folate, homocysteine, methionine-load test, metabolism, methyl donor, S-adenosylmethionine

## Abstract

Methyl groups can be obtained either from the diet (labile methyl groups) or produced endogenously (methylneogenesis) via one-carbon (C1-) metabolism as S-adenosylmethionine (SAM). The essential nutrients folate and choline (through betaine) are metabolically entwined to feed their methyl groups into C1-metabolism. A choline-deficient diet in rats produces a 31–40% reduction in liver folate content, 50% lower hepatic SAM levels, and a doubling of plasma homocysteine. Similarly, folate deficiency results in decreased total hepatic choline. Thus, sufficient intakes of both folate and choline (or betaine) contribute to safeguarding the methyl balance in the body. A significant amount of choline (as phosphatidylcholine) is produced in the liver via the SAM-dependent phosphatidylethanolamine methyltransferase. Experimental studies using diets deficient in several methyl donors have shown that supplemental betaine was able to rescue not only plasma betaine but also plasma folate. Fasting plasma homocysteine concentrations are mainly determined by folate intake or status, while the effect of choline or betaine on fasting plasma homocysteine is minor. This appears to contradict the finding that approximately 50% of cellular SAM is provided via the betaine-homocysteine methyltransferase (BHMT) pathway, which uses dietary choline (after oxidation to betaine) or betaine to convert homocysteine to methionine and then to SAM. However, it has been shown that the relative contribution of choline and betaine to cellular methylation is better reflected by measuring plasma homocysteine after a methionine load test. Choline or betaine supplementation significantly lowers post-methionine load homocysteine, whereas folate supplementation has a minor effect on post-methionine load homocysteine concentrations. This review highlights the interactions between folate and choline and the essentiality of choline as a key player in C1-metabolism. We further address some areas of interest for future work.

## 1. Introduction

Methyl groups are used by numerous methyltransferases that play a role in cellular processes such as DNA methylation or the synthesis of neurotransmitters and amino acids [1,2]. Methyl groups can be obtained either from the diet (labile methyl groups) or produced endogenously (methylneogenesis) via one-carbon (C1-) metabolism [3]. Folate, choline (or lecithin), betaine, and methionine are examples of the dietary sources of labile methyl groups (Figure 1).

The possibility of deriving methyl groups from multiple nutrients allows the body a buffer in safeguarding the methylation reservoir. The requirements for each of the dietary methyl donors are partly determined by the amount of other methyl donors in the diet and additionally depend on factors such as genetic polymorphisms in C1-metabolism and life stage (i.e., during pregnancy and lactation). Conservation of methyl groups in the body is necessary to optimize C1-metabolism [10], meet body requirements, and excrete methylated substrates in the urine. This narrative review highlights the essential role of folate and choline as key players in C1-metabolism and their interactions, and discusses some of the translational aspects of these functional interactions.

## 2. Health Effects of Sufficient Folate and Choline Intakes

Studies in humans have established causal associations between insufficient folate intake and status and several disease conditions. There is high-grade evidence that supplementation of folate beyond the usual dietary intake can reduce the risk of serious birth defects [11] and improve the course of neurodevelopmental [12] and neuropsychiatric disorders [13]. These effects may be explained by the role of folate in DNA synthesis and stabilization, enhancing cellular methylation, and lowering plasma fasting homocysteine concentrations.

In addition to being a source of methyl groups, choline has several other functions in the body, such as building cell membranes (as phosphatidylcholine) and neurotransmission (as acetylcholine). Depletion of dietary choline in adults (50 mg/day) causes fatty liver within 3 weeks, while choline supplementation (i.e., 500 mg/day) normalized liver function [14,15]. Depletion of dietary choline during pregnancy in rodents caused fatty liver to develop towards the end of the pregnancy, not only in the mothers, but also in the fetuses [16]. Therefore, a sufficient choline intake during all life stages, especially during pregnancy and lactation, is necessary for the normal function of the liver. A recent systematic review and meta-analysis of observational studies has shown that a lower maternal choline intake is associated with a higher risk of neural tube defects [17]. This association suggests that choline and folate intakes during pregnancy may have additive or even synergistic preventative effects.

Depletion studies in animals have shown that, on the one hand, folate deficiency causes depletion of liver choline [18] and exaggeration of hepatic steatosis [19]. On the other hand, a diet depleted of choline (or choline and methionine) causes depletion of liver folate [7,8,20]. Therefore, the ability to switch between folate and choline as sources of methyl groups can partly secure cellular methylation and underscores the necessity of supply redundancy to safeguard this critically important methyl pool. The essential nutrients folate and choline are metabolically entwined to feed their methyl groups into the C1-metabolism (Figure 1). People carrying polymorphisms in genes involved in folate or choline metabolism may have higher requirements for both nutrients [21]. Deficiencies of folate and choline show similar (e.g., elevated homocysteine, lowered methylation potential), but also distinct phenotypes. Based on this, it is tempting to assume that higher dietary intakes of both folate and choline (e.g., during the preconception and early pregnancy) can be translated into more effective disease prevention on a population level compared to when only folic acid is supplemented.

## 3. Formation of Methyl Groups in C1-Metabolism

C1-metabolism ensures the disposition of CO_2_ from different sources, such as glucose metabolism, glycine decarboxylation, and 10-formyltetrahydrofolate dehydrogenase into the folate pathway. Additionally, it regenerates adenosine triphosphate (ATP) from adenosine diphosphate (ADP) through conversion of 5-formyltetrahydrofolate (5-formyl-THF) to tetrahydrofolate (THF). C1-metabolism contributes to the synthesis of glutathione from cysteine, and the de novo synthesis of methionine, adenosine, guanosine, and thymidylate. Seventy percent of methionine (a protein-forming amino acid) is obtained from dietary protein degradation. The remaining 30% is formed from homocysteine in a remethylation pathway that contributes to the body’s methyl balance [22]. Serine, formate, glycine, dimethylglycine, and sarcosine introduce C1 units into the 5,10-methylenetetrahydrofolate pool.

### Homocysteine Methylation to Methionine

Homocysteine is remethylated to methionine via methionine synthase (MS) and betaine homocysteine methyltransferase (BHMT). Both enzymes are highly expressed in the liver and the kidney [23,24]. Homocysteine is converted to methionine via MS using the methyl group of 5-methyltetrahydrofolate (5-methyl-THF) in the presence of vitamin B12 (as methylcobalamin). In a subsequent step, methionine adenosyltransferase, an ATP-dependent enzyme, converts methionine into S-adenosylmethionine (SAM). SAM is a universal methyl donor for numerous methyltransferase-dependent cell reactions. S-adenosylhomocysteine (SAH) is formed after transferring a methyl group from SAM to a methyl acceptor (Figure 1). SAH is converted to homocysteine in an irreversible reaction, and homocysteine is either recycled back to methionine to further produce methyl groups or is converted to cystathionine, and finally glutathione [25].

The methyl groups of betaine are used to convert homocysteine to methionine via the BHMT pathway, thus generating SAM [26]. Betaine is obtained either directly from the diet or via an irreversible mitochondrial oxidation of choline mediated by choline dehydrogenase (CHDH) (Figure 1).

Hyperhomocystinuria is an inherited disorder due to the deficiency of either MS or cystathionine beta synthase (converting homocysteine to cystathionine and thereby cysteine), resulting in the accumulation of homocysteine. Intervention with betaine has been used to lower plasma homocysteine in these patients [27], where folic acid supplementation may not be sufficient and homocysteine can be mobilized via the BHMT pathway. In healthy humans, the BHMT pathway is naturally upregulated to remove homocysteine after ingesting a methionine-rich diet, which is the basis of the methionine load test. Massive disruption of the BHMT pathway, such as in Bhmt−/− mice, causes elevated homocysteine levels that cannot be normalized by supplementing folate (i.e., by channeling homocysteine through the MS pathway) [28]. In addition to hyperhomocysteinemia, Bhmt−/− mice show 48% lower liver SAM content, 21-fold-higher liver betaine, and histological evidence of liver steatosis compared to the Bhmt+/+ mice [4]. The majority of the knockout animals also developed liver tumors later in life [4]. Failure to export triacylglycerol from the livers of the Bhmt−/− mice can be explained by markedly decreased levels of choline derivatives such as phosphocholine, phosphatidylcholine (PtdCho), and sphingomyelin [4]. The liver synthesis of choline derivatives relies on the availability of sufficient methyl groups. This might explain the role of adequate betaine and choline intake in removing triglycerides from liver cells [15]. The knockout animal models provided a quantitative estimate of the relative contribution of BHMT in C1-metabolism, although we acknowledge the limitations of directly extrapolating these results to humans.

C1-metabolism is subject to feedback mechanisms, and it is regulated by the relative availability of dietary methyl donors. Inadequate methyl groups, due to prolonged fasting or short-term limited intake of proteins, serine, folate, or choline, cause upregulation of the re-methylation pathways to convert more homocysteine to methionine. While under these conditions, the flow of homocysteine via the transsulfuration pathway to cystathionine is downregulated [22]. Excess SAM inhibits both the BHMT and methylenetetrahydrofolate reductase (MTHFR) enzymes in order to slow down the production of SAM.

Experimental dietary folate deficiency in humans (50 µg total folate/day for 6 weeks) leads to a 115% increase in fasting plasma homocysteine compared to people on a control diet with sufficient folate [9]. The 115% increase in fasting plasma homocysteine under the folate-deficient diet was associated with only a 37% lower remethylation rate of homocysteine to methionine compared to the control diet [9]. Thus, the rise in fasting plasma homocysteine in folate deficiency is not explained by a corresponding decline in folate-mediated homocysteine remethylation. The question is, where could this additional homocysteine come from? The authors of this study argued that under the folate depletion model in the study, the BHMT-mediated remethylation of homocysteine to methionine was not stimulated to compensate for folate deficiency [9]. Instead, it was posited that folate deficiency may cause a reduction in homocysteine flow through the MS pathway, and cause the one-carbon unit of 5,10-methylenetetrahydrofolate to be directed toward serine synthesis, which may save the C1-units for important cellular reactions [9]. However, the study did not control for betaine and choline intakes. It is theoretically possible that the BHMT pathway was upregulated in the livers of folate-deficient people as an alternative source of SAM. The formation of SAH from SAM after methyl group transfer may explain the 115% increase in plasma homocysteine.

Therefore, the BHMT and MS pathways have distinct roles in C1-metabolism, and folate and betaine/choline roles in this pathway are not fully exchangeable. Normal function of the BHMT pathway (including sufficient intake of choline) contributes to normal homocysteine metabolism, provision of methyl groups, lipid transport, and normal liver function.

## 4. Determinants and Indicators of C1-Metabolism

Elevated fasting plasma homocysteine concentrations may indicate disturbed C1-metabolism due to folate deficiency [29]. Concentrations of fasting plasma homocysteine and plasma folate are routinely used to diagnose folate deficiency. The diagnostic utility of SAM, SAH, choline, betaine, or any methylated substrates (such as DNA methylation) has not been established due to inherent analytical and biological challenges.

Increasing folate intake is the most effective measure for lowering fasting plasma homocysteine levels. Betaine and choline can also lower plasma homocysteine, albeit to a lesser extent. A 2-fold-higher choline and betaine intake (383 mg vs. 689 mg/day) was associated with approx. 1 µmol/L lower fasting plasma homocysteine (−8% versus the low intake category) in adults [30]. The association between choline and betaine intake on the one hand and fasting homocysteine on the other hand was most pronounced among women with a low folate intake [30], suggesting enhanced homocysteine remethylation via the BHMT pathway in women with low folate. Supplementing choline (2.6 g per day as phosphatidylcholine for 2 weeks) [31] or betaine (1.5 g, 3 g and 6 g per day for 6 weeks) [32] caused a dose–response, although moderate, reduction of fasting homocysteine compared to the placebo (−1.3 µmol/L to −2.2 µmol/L or −12% to −20%). Supplementing folic acid for 6 weeks lowered fasting homocysteine to a greater extent than betaine (−18% for folic acid vs. −11% for betaine, both compared to placebo) [5]. These results show that folate has a dominant homocysteine-lowering effect compared to betaine and choline.

## 5. The Methionine Load Test: A Functional Test of the BHMT Pathway

This is a functional test that uses the accumulation of homocysteine in the blood after the administration of a bolus load of methionine to determine the capacity of C1-metabolism by the degree of homocysteine clearance. The methionine load test can therefore detect disorders in C1-metabolism not captured by measuring fasting homocysteine. In this test, 75–100 mg methionine per kg body weight (5.6 g to 7.5 g methionine for a 75 kg person) is ingested after a sample for fasting plasma homocysteine is drawn and then measured after 4, 6, 8, or 12 h post methionine load (PML) [33]. Roughly 50% of people with hyperhomocysteinemia after a methionine load may have normal fasting plasma homocysteine results [34,35,36].

Van der Griend et al. defined PML-hyperhomocysteinemia as an increase in homocysteine concentration after a methionine load (6 h min fasting concentration) of >42.3 µmol/L (the 95th percentile of control subjects) [34]. The definition of PML hyperhomocysteinemia varies between studies. A direct comparison of the test results between studies is not possible due to a lack of standardization regarding the time point at which homocysteine is measured after the methionine dose or the need to adjust for baseline homocysteine [33]. Furthermore, the PML-homocysteine concentrations can be influenced by nutritional and genetic factors (Table 1). Nonetheless, a positive PML test can indicate where choline or betaine supplementation could be used to enhance homocysteine removal via the BHMT pathway (Table 1).

Both acute and chronic intakes of either betaine or choline cause a significant reduction of the PML-homocysteine response. A single dose of 1.5 g, 3 g, or 6 g betaine attenuated the increase in plasma homocysteine after a methionine load (by 16%, 23%, and 35%, respectively, compared to the placebo) [32]. Similarly, a single dose of 1.5 g choline (from phosphatidylcholine) lowered the PML homocysteine by 15% (−4.8 µmol/L; 95% CI: −6.8, −2.8 µmol/L) compared to the placebo [31], suggesting that choline was converted to betaine and used as an immediate source of methyl groups. A 32% reduction of PML-homocysteine was also reported after 6 h of a meal that was rich either in betaine or in choline compared to a control meal that was depleted of the two nutrients [6]. Therefore, the PML-homocysteine concentration seems to be a sensitive marker of recent intake of choline and betaine. This test may be used to define the intakes of betaine or choline at the inflection point of the homocysteine curve (i.e., when a higher intake of betaine or choline does not lead to a further reduction in PML-homocysteine).

It has been shown that concentrations of plasma betaine correlate with PML-homocysteine concentrations in general [38,39]. The correlation remained significant when people received supplements containing folic acid [38] or when plasma folate concentrations were accounted for [39]. Thus, PML-homocysteine is a sensitive marker of low plasma betaine (and possibly choline). This suggestion is strengthened by results from a randomized placebo-controlled trial measuring PML-homocysteine levels before and after supplementing either 400 µg × 2 per day folic acid or 3 g betaine × 2 per day for a duration of 6 weeks (both interventions were tested against a placebo) (Table 2) [5]. Betaine supplementation for 6 weeks caused a strong reduction of the PML-homocysteine concentrations compared to the placebo group [mean difference (95%CI) at 6 h = −17.7 (−31.8, −5.3) µmol/L]. Folic acid supplementation caused a non-significant change in the PML-homocysteine compared to the placebo [mean difference (95%CI) at 6 h = 1.1 (−13.2, 15.4) µmol/L] [5] (Table 2). Similarly, supplementation with 2.6 g choline per day (as phosphatidylcholine) for 2 weeks caused 29% lower PML-homocysteine concentrations compared to the placebo [31] (Table 2).

At present, the methionine load test is not widely used in clinical practice due to its complexity and time demands. In the absence of recognized markers of choline and betaine status and intake, the PML-homocysteine can be considered a functional marker of the BHMT pathway. This marker shows the relative contribution of betaine and choline as methyl donors. Available evidence suggests that PML-hyperhomocysteinemia is a functional marker of disturbed C1-metabolism. Combined supplementation of folate and choline or betaine can normalize C1-metabolism, although this effect is not mirrored by lowering fasting plasma homocysteine concentrations but is well reflected by PML homocysteine. Some gaps in knowledge about the potential use of PML homocysteine in diagnostic and research areas are shown in Table 3.

## 6. Safeguarding the Methyl Balance Through Diet or Methylneogenesis

The diet provides a wide intake range of methyl sources. Individuals meeting the current dietary intake recommendations for methyl donors are unlikely to have impaired methylation. However, the methylation equilibrium could be disrupted under conditions of high requirement, increased loss, or low intake of one or more methyl donors. De novo formation of new methyl groups (methylneogenesis) can maintain the net amount of methyl groups in the body when dietary intake of methyl donors is temporarily limited [50] and during prolonged fasting conditions.

The body methylation balance is influenced by renal excretion of SAM and other methylated compounds such as creatine, creatinine, *N*-methyl nicotinic acid, carnitine, methylated amino acids, and the terminal oxidation of methyl groups, including that of sarcosine (or *N*-methylglycine) [51]. The relative contribution from the loss of methyl groups in the bile and stool to body methyl balance remains unknown. In addition, biliary excretion of PtdCho was estimated to consume 5 mmol SAM per day, but it is unknown whether this PtdCho is reabsorbed.

There have been some attempts to quantify body methyl flux, which reflects the overall body dynamics of methyl groups in µmol·kg^−1^·h^−1^. Using the stable isotope infusion technique, the daily methyl flux in healthy subjects was estimated to be between 16.7 and 23.4 mmol/L (for a 70 kg person) when combining fasting and fed situations within a 24-h period [50]. Approximately 40% of the daily formed homocysteine is remethylated to methionine, and 54% of the methyl groups required for this remethylation were generated via the methylneogenic pathway [52]. Homocysteine remethylation using the methyl group of 5-methyl-THF has been estimated to be between 2 and 8 µmol·kg^−1^·h^−1^ [52], suggesting that homocysteine remethylation via methionine synthase constitutes a small fraction of human one-carbon metabolic flux. Considering that Bhmt knockout mice had 48% lower liver SAM compared to the controls, it can be inferred that roughly 50% of SAM is derived from the BHMT pathway [4].

Folate and choline are also key sources of one-carbon units that feed into the folate cycle [50]. Sarcosine can be synthesized from SAM-dependent methylation of glycine or from oxidation of choline to betaine, which is converted to dimethylglycine and then to sarcosine. A substantial amount of one-carbon units originates from glycine and serine [53]. Methionine loading (such as after a protein-rich meal) stimulates the activities of some methyltransferases, such as guanidoacetic methyltransferase [54] and glycine *N*-methyltransferase [51], which can lead to the formation of creatine and sarcosine, respectively.

## 7. Factors Affecting Methylneogenesis

C1-metabolism is influenced by the availability of one-carbon moieties in the diet, sex, age, genetic polymorphisms, and life stage (e.g., pregnancy, lactation, and early life). Some of these factors have been extensively investigated.

The activities of several enzymes involved in C1-metabolism differ by sex [48]. The expression of the phosphatidylethanolamine methyltransferase (PEMT) gene is responsive to estrogen [55,56]. Therefore, the reliance of PEMT on methyl groups is high in premenopausal women and highest during the third trimester of pregnancy. PEMT activity may be lowest in postmenopausal women and men, suggesting that choline requirements could be higher in those subgroups. Moreover, people with a homozygous PEMT SNP [57,58] are more sensitive to developing symptoms of choline deficiency (e.g., fatty liver or muscle damage) than people without this SNP, especially under restricted choline and/or folate intake.

The presence of a homozygous variant of the common polymorphism in the MTHFR gene (C677T: TT genotype) causes lower activity of the enzyme and results in lower circulating concentrations of folate. Mthfr−/− mice have a high incidence of postnatal death [59], hyperhomocysteinemia, hypomethylation, and are prone to develop fatty liver under choline-deficient diet conditions [60]. Betaine supplementation to pregnant mice throughout pregnancy and until weaning of the pups at 3 weeks of age decreased the mortality of Mthfr−/− mice from 83% to 26%, lowered plasma homocysteine, and increased methionine and SAM concentrations in the liver and brain [59], showing that the BHMT pathway can rescue at least part of the severe metabolic and phenotypic consequences of MTHFR deficiency.

In humans, the MTHFR677TT genotype is associated with elevated fasting plasma homocysteine levels, low serum and blood folate concentrations [61,62], and an attenuated response to folate supplementation [61]. Yan et al. showed that the demand for betaine to generate methionine from homocysteine is likely higher in individuals with the 677TT genotype [62]. In a 12-week study of 60 men with a known MTHFRC677 T genotype, participants received daily supplementation with 550 mg or 1100 mg choline for 9 weeks, followed by d-9 labeled choline for an additional 3 weeks [62]. The higher ratio of d-9-betaine to d-9-PtdCho in the 1100 mg choline group vs. the 550 mg group indicated greater shunting away from the CDP-pathway and down the PEMT pathway [62]. The higher plasma betaine/CDP-choline ratio in subjects with the TT genotype is consistent with increased demand for choline and, therefore, the need to channel a higher proportion of choline into betaine under limited folate status.

The above discussed factors that influence methylneogenesis can significantly impact personalized intake of methyl donor nutrients in some populations or population subgroups. To date, population intake recommendations for choline and folate have taken some of these considerations into account. For example, a higher intake of folate and choline is recommended during pregnancy and lactation than outside this period [63,64]. The United States National Academy of Medicine (NAM) has upgraded choline intake recommendations for men compared to women (+50 mg/day higher intake recommendation for men) [65]. The European Food Safety Authority (EFSA) has upgraded population intake recommendations for folate by 15% due to prevalent MTHFR C677T polymorphism where people require higher folate intake to achieve normal folate status [63]. In contrast, polymorphisms in the PEMT and BHMT genes (e.g., rs1316753 and rs1915706) were not taken into consideration when setting the intake recommendations for folate and choline; however, evidence suggests that this issue warrants reassessment.

## 8. The Role of Phosphatidylethanolamine Methyltransferase in Methylneogenesis

The enzyme PEMT is a SAM-dependent enzyme that is responsible for the synthesis of 30% of PtdCho in the liver [66,67]. PEMT utilizes three SAM molecules to convert one molecule of phosphatidylethanolamine into PtdCho [68], and as a result, three molecules of SAH and consequently homocysteine are produced. One of the three methyl groups of PtdCho originates from homocysteine remethylation via the BHMT pathway.

When deuterium-labeled choline is used and the methyl groups are tracked, it was found that the choline diphosphate (CDP) deuterated methyl groups are detected as d-9-PtdCho, showing that the labelled methyl group is channeled down the Kennedy pathway. When choline is driven down the PEMT pathway via methionine, then only one deuterated methyl group can be detected as d-3-PtdCho (Figure 2). PtdCho derived from the PEMT pathway requires three sequential methylation reactions of phosphotidylethanolamine (PE), which generate d-3-PtdCho, d-6-PtdCho (if it has two methyl groups from the originally labeled choline or betaine), or rarely d-9 (three deuterium-labeled methyl groups).

Metabolites can also be identified by this method (d-6-dimethylglycine, d-3-sarcosine, d-3-methionine, and d-3-SAM). Figure 2 illustrates the metabolic fate of choline’s methyl groups as demonstrated by compartmentalization studies [65].

The PEMT pathway is a significant contributor to methyl group homeostasis [50]. It has been estimated that PEMT-mediated PtdCho synthesis may consume 5 mmol SAM per day [69], thus implying that an equivalent amount of SAH and homocysteine is produced. Stead et al. reported that mice lacking PEMT activity have 50% lower plasma homocysteine and suggested that the PEMT pathway is a significant source of homocysteine and, therefore, SAM in the body [69]. Independent experiments have shown that hepatocytes isolated from PEMT−/− mice have lower PEMT activity and secrete less homocysteine than hepatocytes from control mice [70]. In contrast, studies on CTP: phosphocholine cytidylyltransferase-(CT) knockout mice, where higher PEMT flux leads to higher PtdCho production, the concentrations of plasma homocysteine were 20–40% higher than in the control mice [71]. Seventy percent of liver PtdCho is produced via the CDP-choline pathway. In CTP: phosphocholine cytidylyltransferase-(CT) knockout mice [71], hepatic activity of the CDP-choline pathway is 80% lower than in control mice, while PEMT mRNA, protein, and activity are upregulated, possibly to compensate for the absence of the CDP-pathway and to produce more PtdCho under the same standard diet [72]. Additionally, the BHMT activity was increased by 80% in the CT knockout mice, thereby securing a source of methyl groups required for PEMT activity through the oxidation of choline to betaine. Therefore, the need to maintain a sufficient amount of PtdCho in this genetic mouse model caused compensatory induction of PEMT and BHMT pathways of the same magnitude [71]. Although BHMT-induction does not lead to normalizing plasma homocysteine, it may maintain SAM at control levels [71]. The compensatory induction of BHMT is similar to the situation in Mthfr^−/−^ and Cbs^−/−^ mice models, where homocysteine accumulates [73,74]. On the other hand, deletion of the BHMT gene in mice has been shown to cause an 8-fold increase in plasma homocysteine, a massive disturbance in hepatic methylation potential (low SAM and high SAH), and accumulation of fats in the liver due to the inability to synthesize sufficient PtdCho to export triglycerides from hepatic cells [4]. Therefore, the BHMT pathway, which relies on adequate intake of choline and betaine, is a significant source of methyl groups needed to produce PtdCho via PEMT. The PEMT pathway consumes SAM, but also provides SAH and thereby homocysteine, which is recycled to methionine and new methyl groups.

PtdCho, which is produced via the PEMT pathway using the methyl groups of folate, betaine, and choline, transports long-chain polyunsaturated fatty acids (PUFAs), such as docosahexaenoic acid (DHA) (22:6*n*-3) [75], a fatty acid highly enriched in the brain and retina. In contrast, phospholipids derived through the CDP pathway carry medium-chain, monounsaturated fatty acids, such as oleic acid [76].

## 9. Tracking the Methyl Groups of Betaine and Choline

The transmethylation reaction via BHMT provides one methyl group to convert homocysteine to methionine, while retaining the remaining two methyl groups of betaine that form dimethylglycine. One methyl group of dimethylglycine provides a C1-unit that is used to convert THF to 5,10-methylene-THF, and sarcosine is produced. Sarcosine, or *N*-methylglycine, carries the last methyl group of dimethylglycine, which is used to convert THF to 5,10-methylene-THF and produce glycine [77]. Thus, one of choline’s methyl groups is used to produce SAM, and the remaining two methyl groups reenter the C1-metabolism pool as 5,10-methylene-THF via formaldehyde. Choline enriches the folate pathway with two carbon units that can join C1-metabolism as labile methyl groups via 5-methyl-THF (Figure 3).

The role of choline in enriching cellular folate has been demonstrated in a study on newborn pigs [78]. The animals were fed a methyl-deficient diet (without folate, choline, and betaine) from 4 to 8 days of age for 7 days and then received a rescue treatment with either folate or betaine between days 7 and 10 [78]. The folic acid group showed normalization of plasma folate, a decline in plasma homocysteine, and no change in plasma betaine [78]. In contrast, in the betaine rescue group, which also showed correction of plasma betaine, there was a significant increase in plasma folate, demonstrating that folate was endogenously produced after the conversion of betaine to sarcosine (Figure 3).

## 10. Interdependency of Folate and Choline

It has also been previously reported that 60% of choline is converted to betaine in the liver. This was confirmed in an in vitro study by DeLong et al., in which the levels of betaine were found to be three times higher than those of choline in normal hepatocyte culture [67]. When hepatocytes were exposed to isotope-labelled choline (d9-choline), the ratio of d9-labeled betaine to choline was the same as the ratio of unlabeled betaine to choline. When choline is absent from the cell system, no betaine can be detected, demonstrating that oxidation of choline is an obligatory source of betaine in hepatocytes [67]. Liver cancer cells (RH7777 hepatoma) do not have the enzyme system to convert choline to betaine. Hence, the 3:1 ratio of betaine to choline is extinguished, thus confirming the significant contribution of choline’s methyl groups to betaine in the liver [67].

In the presence of a choline/betaine deficiency, there is greater dependence on 5-methyl-THF to generate PtdCho via the PEMT pathway (Figure 2). A choline-deficient diet for 2 weeks in rats caused a 31–40% reduction in liver folate content, which was reversible when choline refeeding occurred [7,8]. Rats fed diets deficient in both methionine and choline for 5 weeks had hepatic folate concentrations that were 50% of those in controls [8]. Tetrahydrofolate deficiency, induced by methotrexate [79,80,81,82,83] or by dietary folate deficiency [18], resulted in decreased hepatic total choline, with the greatest decrease occurring in hepatic phosphocholine concentrations. During choline deficiency, hepatic SAM concentrations also decreased by as much as 50% [84,85,86,87].

These results suggest that temporary low intake of one of these nutrients, such as due to seasonal food availability or food choice, can be compensated for by other nutrients or methylneogenesis in C1-metabolism. However, one-sided nutrition or a lack of methyl donors during critical stages of the life cycle, such as pregnancy, lactation, and early life, can have serious consequences for health.

## 11. Conclusions

The essential nutrients, folate and choline, are metabolically entwined to provide SAM for functional methylation reactions. A folate-deficient diet depletes liver choline, and a choline-deficient diet depletes liver folate by up to 40%, suggesting that long-term depletion of one of these nutrients can have devastating effects on cellular methylation capacity. Dietary folate intake or plasma concentrations of 5-Methyl-THF play a dominant role in determining fasting plasma homocysteine concentrations, but this role is not exclusive. Intakes of choline and betaine show weak associations with fasting plasma homocysteine. Studies on Bhmt−/− mice suggest that this choline- or betaine-dependent enzyme contributes to 50% of cellular SAM. Still, its role in methylation is not reflected by measuring fasting plasma homocysteine, nor by the effect of supplemental betaine or choline on lowering fasting homocysteine. Instead, the BHMT pathway’s major contribution to removing homocysteine can only be appreciated after a methionine load test (100 mg methionine/kg body weight). At the same time, folate has a limited effect on post-methionine hyperhomocysteinemia. Measuring PML homocysteine concentrations can be used as a sensitive marker of the contribution of choline and betaine to cellular methylation. The multiple scientific studies reviewed here support a significant and so-far underestimated role of choline and betaine as sources of methyl groups.

## Figures and Tables

**Figure 1 nutrients-17-02495-f001:**
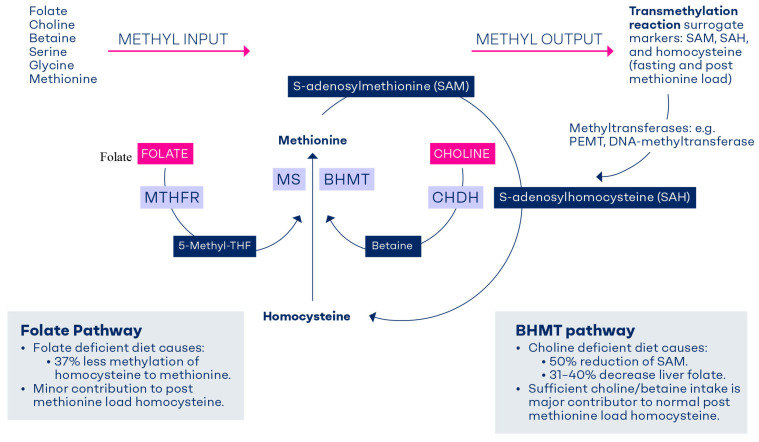
Flux of methyl groups of folate and choline. Coordination between the betaine homocysteine methyl transferase (BHMT) and folate pathways (methionine synthase, MS) ensures balanced methylation. The relative contribution of the BHMT pathway versus the folate pathway to the methylation balance may vary according to the marker used to capture cellular methylation. The BHMT pathway provides 50% of the methionine or SAM [4] and is a major contributor to removing homocysteine after a methionine load [5,6]. A choline-deficient diet lowers liver folate by 31–40% [7,8]. Folate deficiency causes 37% lower methylation of homocysteine to methionine (thus elevated fasting homocysteine) [9], but has a limited effect on post-methionine hyperhomocysteinemia [5,9]. CHDH, choline dehydrogenase; MTHFR, methylenetetrahydrofolate reductase; PEMT, phosphatidylethanolamine methyl transferase.

**Figure 2 nutrients-17-02495-f002:**
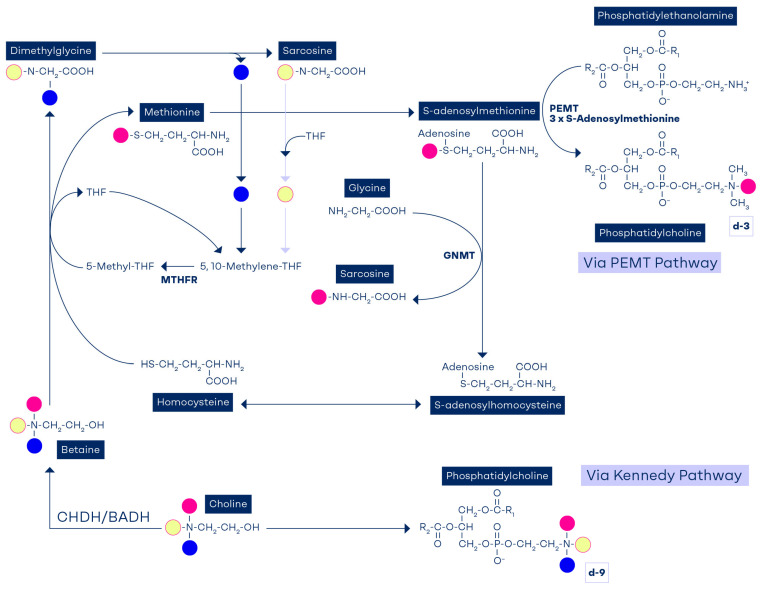
Metabolic fate of orally consumed deuterium-labelled choline. The methyl groups of choline are marked with different colors. The d-9-choline tracer contained three deuterium-labelled methyl groups. After irreversible oxidation of choline to betaine by choline dehydrogenase (CHDH) and betaine aldehyde dehydrogenases (BADH), one of the three labelled methyl groups is used to convert homocysteine to methionine and appears in the S-adenosylmethionine molecule. The second labelled methyl group (now on dimethylglycine) is traced in 5,10-methylene-tetrahydrofolate. The third methyl group of choline is channeled via sarcosine and then transferred to tetrahydrofolate, which is traced in 5,10-methylene-tetrahydrofolate. Thus, each choline molecule that is oxidized to betaine contributes three methyl groups to the methyl reservoir. The synthesis of phosphatidylcholine from phosphatidylethanolamine via phosphatidylethanolamine methyltransferase (PEMT) utilizes three methyl groups from the methylation pool. Also, it generates S-adenosylhomocysteine, which feeds back into the homocysteine remethylation pathway.

**Figure 3 nutrients-17-02495-f003:**
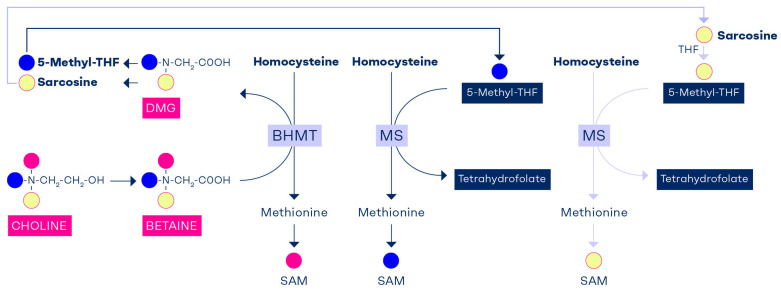
The methyl groups of choline and betaine are highly conserved and used to replenish the methyl reservoir. The methyl groups of choline are marked with different colors. Each choline molecule that is oxidized to betaine can theoretically convert three homocysteine molecules to form three S-adenosylmethionine (SAM) molecules; one SAM is formed directly from betaine via the BHMT pathway, while the other two SAMs are synthesized through the formation of 5-methyl-THF (one from dimethylglycine and the other from sarcosine). The two 5-methyl-THF molecules then augment SAM via the methionine synthase pathway. This could explain why betaine supplementation causes a rise in plasma folate in a methyl-deficient animal model [78].

**Table 1 nutrients-17-02495-t001:** Determinants and conditions of homocysteine concentrations after methionine load.

Determinants of Homocysteine Concentrations After Methionine Load	Direction
Elevated fasting plasma homocysteine [37,38].	↑↑
Higher plasma betaine [37,38,39]; betaine intake (diet or supplements) [5,6,40]; or choline intake (diet or supplements) [6,40,41].	↓ ↓ ↓
Acute intake of choline/betaine (single dose studies or after a meal) [31,32].	↓ ↓ ↓
Higher intake of serine or cysteine [42]; higher folate status [39]; or folate intake [5].	↓
Polymorphisms in the transsulfuration pathway (e.g., cystathionine β-synthase) [43,44].	(↓↑)
Low vitamin B12 status [45]; vitamin B6 supplementation [38,46].	(↑); (↓)
**Conditions Where the Post-Methionine Load Test May Be Used to Detect Hyperhomocysteinemia**	
Insufficient choline intake or status.	
Carriers of polymorphisms in methylenetetrahydrofolate reductase (MTHFR) [47], phosphatidylethanolamine methyl transferase (PEMT) or BHMT genes.	
Anti-folate drugs (e.g., methotrexate, antimalarial) or drugs interfering with folate absorption/metabolism.	
Pregnant and lactating women and children with high choline requirements not met through diet.	
B12 deficiency (e.g., vegan, elderly).	
Mild to moderate fasting hyperhomocysteinemia not explained by low folate, B6 or B12 concentrations.	

↑↑ higher, ↓ ↓ ↓ markedly lower, ↓ lower, (↑) slightly higher, (↓) slightly lower, (↓↑) no clear effect.

**Table 2 nutrients-17-02495-t002:** Relationship between post-methionine load (PML) and fasting (F)-homocysteine levels under different intake conditions.

PML-Homocysteine	F-Homocysteine	PML-Minus F-Homocysteine	%Change (PML vs. F-Homocysteine)	Prediction of PML-Homocysteine From F-Homocysteine Under Different Intake Conditions
**1: Native Condition—No Supplement (Olthof et al. [31] and Steenge et al. [5])**				
32.6 µmol/L	15.6 µmol/L	17.0 µmol/L	+109.0%	PML-homocysteine = F-homocysteine × 2.09.
34.8 µmol/L	12.2 µmol/L	22.6 µmol/L	+185.2%	PML-homocysteine = F-homocysteine × 2.85.
31.6 µmol/L	13.0 µmol/L	18.6 µmol/L	+143.1%	PML-homocysteine = F-homocysteine × 2.43.
** *Mean = 33.0 µmol/L* ^1^ **	** *13.6 µmol/L* **	** *19.4 µmol/L* **	** *+145.8%* **	***PML-homocysteine = F-homocysteine*** × ***2.46.***
**2: Supplemented with 2.6 g/d Choline for 2 Weeks (Olthof et al. [31])**				
22.3 µmol/L	13.6 µmol/L	8.7 µmol/L	+64.0%	PML-homocysteine = F-homocysteine × 1.64. Methionine load test in choline intake-optimized persons led to roughly 55% lower PML-homocysteine compared to non-supplemented people (8.7 vs. 19.4 µmol/L)
**3: Supplemented with 3** × **2 g/d Betaine for 6 Weeks (Steenge et al. [5])**				
17.6 µmol/L	10.9 µmol/L	6.7 µmol/L	+61.5%	PML-homocysteine = F-homocysteine × 1.62. Methionine load test in betaine intake optimized persons led to roughly 65% lower PML-homocysteine compared to non-supplemented people (6.7 vs. 19.4 µmol/L)
**4: Supplemented with 400 µg** × **2/d Folic Acid for 6 Weeks (Steenge et al. [5])**				
33.0 µmol/L	10.7 µmol/L	22.3 µmol/L	+208.4%	PML-homocysteine = F-homocysteine × 3.1. Optimization of folate status has no lowering effect on PML-homocysteine compared to non-supplemented people (22.3 vs. 19.4 µmol/L)

^1^ italics indicate the mean value of the results in the first three lines.

**Table 3 nutrients-17-02495-t003:** Gaps in knowledge surrounding the use of PML-homocysteine to capture disorders in methyl group flux through the BHMT pathway (or alternatively, choline and betaine).

Question	Elaboration
Sex differences: Is the methylation flux higher in men than in women?	The expression of several enzymes in C1-metabolism show sex differences [48]. For example, men have higher plasma homocysteine and betaine than premenopausal women because the PEMT gene is upregulated by estrogen.
Is there a dose–response relationship between choline intake and PML-homocysteine?	A dose–response relationship between betaine intake and PML-homocysteine has been demonstrated [32]. Does the same apply for choline and what is the intake level of choline to achieve a maximal reduction of PML-homocysteine?
Could high dose choline or betaine compensate for folate deficiency in terms of lowering PML-homocysteine?	Addressing metabolic capacity to upregulate methyl group flow via betaine/choline in individuals with folate deficiency or MTHFRC677T TT genotype.
Can PML-homocysteine be used to define the optimal intake of choline or betaine in pregnant and lactating women?	Homocysteine concentrations after a methionine load test can be measured before and after correcting the ‘gap’ of choline or betaine intake.
Can the PML-homocysteine test be used to identify women at high risk of neural tube defects or other pregnancy complications such as recurrent pregnancy loss, gestational diabetes, or preeclampsia?	In one study among women with a history of recurrent pregnancy loss, folic acid supplementation (0.5 mg/d for 2 months) did not lower PML-homocysteine in 53% of the women [49]. In theory, the PML-homocysteine test may identify women who could benefit from choline/betaine supplements by increasing methyl group flux via the BHMT pathway and normalizing PML-homocysteine. This may influence disease risk.

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
