# Peer review of "The Shuttling of Methyl Groups Between Folate and Choline Pathways"

_nutrients, 2025, doi:10.3390/nu17152495_

Round 1
Reviewer 1 Report
Comments and Suggestions for Authors
Suggestion for authors of nutrients-3763471_Review
Title: The shuttling of methyl groups between folate and choline pathways
Description of the manuscript content: The authors discussing the role and relationship between the folate and choline/betaine cycle and their contribution to C1-metabolism in 13 pages with 3 figures and 5 tables. The text is well written, easy to understand, well structured. The topic is interesting and worth publishing, although can be improved in some way.
General comments: Please check the format requirements for maximum figure+table amount. I think 6 is the maximum number. If possible please put those which can be presented together. Although the references are adequate they are quite old. From the 77 ref only around 10 which id after 2010. It could be accepted if the topic is not investigated that much, however in the scientific literature more publication is out from 2005 than earlier.
Detailed suggestion:
- I miss references in the Introduction section. At least for sentences ending in line 45 and 61. But can be added to those ending in line 41, 43, 55, 57 and 58 as well.
- The Figure legend from line 66 “The BHMT pathway etc…” should be placed into the main text of the introduction and cite the Figure 1.
- In Figure 1 and 2 please add SAM, SAH abbreviation to the figure and legend after their full name. You use in the text several times, thus easier to follow the meanings if the abbrev is also there.
- Please add to Figure 1. CHDH onto the line connecting choline and betaine, and similarly MTHFR onto the line connecting folate and 5-Methyl-THF. Reference the Fig1 in the relevant sentence in section 2.
- Please put closer the MS and BHMT to the arrow connecting homocysteine with methionine in Figure 1. to be more visible where they act. Reference the Fig1 in the relevant sentence in section 2.
- Should expand the part of Introduction section from line 52-61 with more recent references and discussion a bit more details of the disease, which related with folate, choline and betaine deficiency (i.e after sentence in line 54-55, 55-57 and 57-61).
- Should create from the altogether 8 (Figures+Tables) somehow 6 without reducing the meaning and content. I think all the information should be in the main text, not in supplementary.

Reviewer 2 Report
Comments and Suggestions for Authors
Summary:
The metabolic interaction between the folate and the choline pathways is discussed with special reference to the importance of transfer of the methyl group in the balance of cellular methylation. It stresses the metabolic interrelationship, the human health implications, and clinical indicators such as post-methionine loading homocysteine levels to determine nutritional adequacy.
Strengths:
-
The manuscript clearly elucidates the biochemical interactions between folate, choline, and betaine and presents a coherent integration of cumulative experimental and clinical evidence.
-
They effectively integrate detailed metabolic pathways into manageable concepts, making it accessible to readers with limited experience with one-carbon metabolism.
-
The inclusion of detailed and schematic diagrams is very helpful in visualizing and understanding the biochemical interactions.
-
The article critically evaluates the efficacy of the methionine loading test and reveals its potential application in clinical diagnostic work, bringing practical value to the theoretical discussion.
Weaknesses:
-
There are areas that are overly detailed and may confuse the student; condensing or rewriting text under clearly labeled subsections may improve readability.
-
The discussion on the role of genetic polymorphisms (e.g., PEMT and MTHFR) and their clinical relevance, although informative, is disjointed and might benefit from better contextualization and integration.
-
The paper concisely reviews animal models (e.g., Bhmt knockout mice) but does not clearly explain the translational significance or limitations of the models as applied to human research.
-
There were slight grammar and language variations that had to be corrected to attain clarity and formality.
Minor Comments:
-
It is best to abbreviate consistently throughout the paper (i.e., first clearly explain PEMT, BHMT, SAM, SAH, etc.).
-
Correction and potential refinement of language:
-
Abstract line 10: "Methyl groups are either obtained from the diet." may run better as "Methyl groups can be obtained either from the diet."
-
Page 3, line 133: “transsulfuation” is spelled incorrectly and must be changed.
-
Page 6 Table 2: Make the table more readable by keeping the numerical data properly aligned and including units consistently in all numbers.
-
-
Figures (specifically, Figure 2 and Figure 3) are excellently descriptive but might benefit from brief explanatory captions which directly refer the figures to corresponding arguments developed in the main text.
Recommendation:
It is an exemplary paper that is systematically reviewing key biochemical interactions between the folate and the choline pathways. Its merits outweigh its limitations significantly. I therefore agree with acceptance with minimal correction, primarily to improve readability, provide better background on genetic polymorphisms, increase linguistic precision, and fortify translational relevance with clinical significance in man.
Comments on the Quality of English LanguageThe English could be improved.
Round 2
Reviewer 1 Report
Comments and Suggestions for Authors
nutrients-3763471_Review
General comment: The manuscript has been improved both in its content and structure. Now 3 figures and 3 tables are in the manuscript of 15 pages (without references). More easy to read now the text and follow the logic.
I have found some part, which still would be beneficial to add and clear our out (please see below). Also I have not noticed in the first version that „adenine” was used instead of „adenosine” – please correct it (see below).
Suggestion:
nutrients-3763471_Review
General comment: The manuscript has been improved both in its content and structure. Now 3 figures and 3 tables are in the manuscript of 15 pages (without references). More easy to read now the text and follow the logic.
I have found some part, which still would be beneficial to add and clear our out (please see below). Also I have not noticed in the first version that „adenine” was used instead of „adenosine” – please correct it (see below).
Suggestion:
1. Line 93: „Isolated deficiencies of folate and choline” - do you, mean „genetic alterations resulted in deficiencies in folate and choline”? If yes, please correct the meaning of the sentence. Recently does not really understandable for me what does the term "isolated def" refer to.
2. Line 94-96. There are results suggesting the high folate can induce tumorigenesis. Need to discuss the possibility of negative effect in terms of too high intake. Which interval (from ?- up to ?) is suggested beneficial?
3. Line 101: ADP, ATP are adenosine diphosphate/triphosphate not adenine di/triphosphate, please correct
Do not need to be sent back these changes, just please be added and corrected. Thank you.
